# Out-Plane Buckling of Arches with Variable Cross-Section

**Angfeng Jiang [1], Deyuan Deng [2,3], Wei Dai [3], Xiuwen You [3] and Hanwen Lu [1,\*]**

[1]  School of Transportation and Civil Engineering & Architecture, Foshan University, Foshan 528225, China; angfengjiang@fosu.edu.cn

[2]  School of Civil Engineering, Southeast University, Nanjing 211189, China

[3]  China Construction Steel Structure Guangdong Corp., Ltd., Huizhou 516200, China; xiuwenyou@fosu.edu.cn (X.Y.)

\*  Correspondence: luhanwen@fosu.edu.cn

**Abstract:** The variable cross-section arch is widely used in practical engineering because of its beautiful arc and excellent mechanical properties. However, there is still no systematic and comprehensive study on the out-plane buckling of variable cross-section arches. In view of this, this paper is focused on the elastic analytical research of out-plane buckling of arches with variable cross-sections under a uniformly distributed radial local load. The pre-buckling and out-plane buckling behavior of a variable cross-sectional arch under an external load is quite different from that of an arch with a uniform cross-section. Castigliano's second theorem is used to establish pre-buckling force method equilibrium equations for variable cross-sectional arches under a uniformly distributed radial local load, and corresponding analytical solutions of normal stress, axial compression, and the bending moments are obtained. Based on the energy method and the Ritz method, analytical solutions of the critical load for the elastic out-plane buckling of arches with variable cross-sections are derived. Comparisons with ANSYS results indicated that the analytical solutions are able to accurately predict the pre-buckling internal forces and critical out-plane buckling load of variable cross-section arches subjected to a uniformly distributed radial local load. It is found that the internal forces and the out-plane buckling load of an arch are significantly affected by the variation of cross-sectional height. As the ratio of the arch's cross-sectional height increases, the bending moment decreases, and the axial force and critical out-plane buckling load increase. Analytical solutions of pre-buckling internal force and critical out-plane buckling load problems for arches with variable cross-sections have a wider significance since they can provide an effective explicit analytic method for the optimal design of arch structures.

**Keywords:** variable cross-section; arch; critical out-plane buckling load; uniformly distributed radial local load

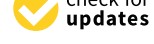



## 1. Introduction

In the design of arch structures, the form of variable cross-section is widely used, and the elastic internal force and critical out-plane buckling load are essential to derive modified slenderness for out-plane stability design. Most of the previous studies focused on uniform cross-section arches for their out-plane buckling. For various load patterns, the out-plane buckling behavior of arch structures has been investigated by several researchers. Trahair [1], Liu et al. [2], and Yuan et al. [3] examined the out-plane buckling of arches, which were subjected to uniformly distributed loads. Pi et al. [4] focused on investigating the out-plane buckling of arches under a central concentrated load. Liu and Lu et al. [5,6] studied the out-plane lateral and torsional instability of arches under arbitrary radial concentrated loads and a central radial point load. In order to better simulate the load patterns observed in actual engineering scenarios, Lu et al. [7] investigated the flexural-torsional buckling of steel arches subjected to a localized uniform radial load. For various constraint boundaries, Pi and Bradford [8] introduced a design equation that predicts the

out-plane strengths of fixed steel arches. Guo et al. [9] developed analytical solutions for the out-plane buckling load of arches with elastic end restraints. Additionally, Xi et al. [10] conducted a buckling analysis of pin-end arches. For various materials, Pi and Trahair [11] conducted theoretical derivations and finite element research on the out-plane inelastic buckling and strength of steel arches. Bouras and Vrcelj [12] investigated the out-plane buckling behavior of concrete-filled steel tubular arches, which are studied in elevated temperatures. Liu et al. [13] explored the out-plane buckling of functionally graded porous I-shaped circular arches with graphene platelet reinforcements. For various cross-sections, Malekzadeh and Karami [14] conducted an out-plane static analysis of an arch with a circular cross-section. Lim and Kang [15] investigated the out-plane buckling of I-shaped arches with a single symmetric axis cross-section, while Xi et al. [16] studied the out-plane buckling of I-shaped arches with a double symmetric axis cross-section. Wang et al. [17] examined the out-plane buckling and in-plane buckling of an arch with a box cross-section. Guo et al. [18] developed methods for determining the out-plane inelastic strength of trussed arches with a rectangular lattice cross-section. Furthermore, Zhong et al. [19] provided theoretical solutions and experimental studies on a rectangular cross-section arch. In addition to the aforementioned studies, there have been numerous investigations on the out-plane buckling of arches, considering various factors such as shearing effects [20], temperature fields [12], creep [21,22], and functionally graded materials [23,24]. However, due to the scope of this paper, not all of these studies will be introduced.

From the existing literature, it is evident that arches with uniform cross-sections have been extensively studied, while only a limited number of studies have focused on arches with variable cross-sections. Huang et al. [25,26] investigated the in-plane vibration of circular arches, which have variable curvature and variable cross-sections. Shin et al. [27] optimized the vibration analysis of circular arches with variable cross-sections. Tsiatas and Babouskos [28] studied the linear and nonlinear responses of non-uniform shallow arches subjected to concentrated forces and developed corresponding integral equation solutions. More recently, Yan et al. [29,30] conducted an in-depth study on the buckling behavior of non-uniform arches divided into three regions with constant stiffness. However, there is a lack of research in the literature concerning the out-plane buckling of arches with variable cross-sections under locally uniform radial loads.

In order to further investigate the elastic analysis of out-plane buckling of a variable cross-section arch under a uniformly distributed radial local load and to fill the gap in the research on variable-section arch under a uniformly distributed radial local load, a novel exponential function variable-section form based on the natural constant *e* is adopted in this paper. Furthermore, this paper combines the Timoshenko beam theory and Castigliano's second theorem to solve normal stress, axial compression, and bending moment of the variable cross-sectional arch before buckling. Based on the precise axial compression and bending moment, the analytical solution of the critical out-plane buckling load of the variable cross-sectional arch is derived, and the numerical model was obtained with ANSYS 18.0 to verify the accuracy of the analytical solution. It provides an effective explicit analysis method for the optimal design of an arch structure. After verifying the accuracy of the internal force and the out-plane critical buckling load of the variable section arch, this article conducted a detailed parametric analysis [31,32]. It is shown that properly designed variable cross-section arches exhibit a more uniform distribution of internal forces and superior buckling performance.

## 2. Basic Assumptions and Cross-Sectional Features

For the arch with variable cross-section, its geometric structure is shown in Figure 1, where $S$ is the arc length, $L$ is the span of the arch, $R$ is the radius of the arch, $f$ is the sagittal height of an arch, $q$ is the locally uniformly distributed radial load, $k$ is the stiffness of the arch end connection, $2\Theta$ is the included angle of the arch, $\theta$ is the twist angle of the cross-section, $c$ is the load area distribution angle, $\phi$ is the angular coordinates of the

arch, $u$, $v$, and $w$ are, respectively, the lateral displacement, radial displacement, and axial displacement.

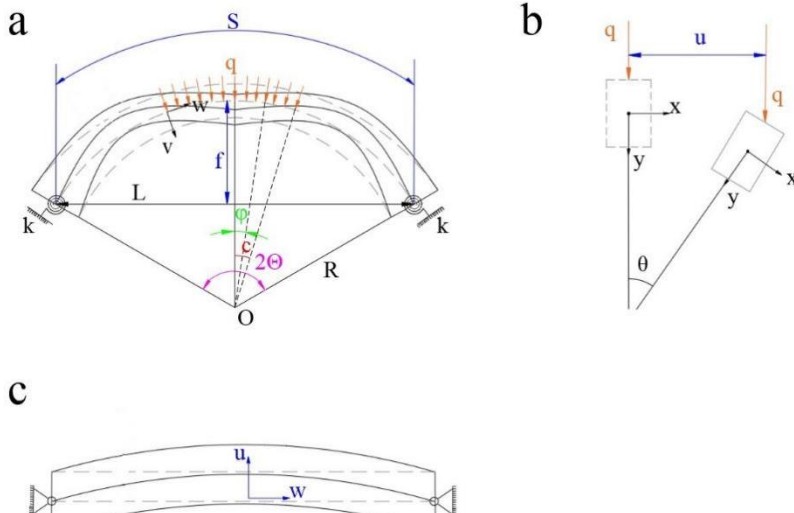

**Figure 1.** Geometric structure of variable cross-sectional arch: (**a**) front view; (**b**) lateral view; (**c**) flat view.

To facilitate the analysis of out-plane buckling in arches with variable cross-sections, the following fundamental assumptions are adopted in this study:

(1) The cross-section of the arch remains perpendicular to the arch axis throughout the buckling process.
(2) An exponential function with the base of the natural constant $e$ is employed as the variable cross-sectional form.
(3) The span and height of the arch are significantly larger than the dimensions of the arch's cross-section are assumed.
(4) The cross-section of the arch is assumed to be rectangular, exhibiting continuous and uniform variations along its length.
(5) The material of the arch is assumed to be uniform, isotropic, and exhibits linear elasticity. The elastic modulus is represented as $E$, and the shear modulus as $G$.

The above assumptions are set up in consideration of the safety of engineering applications. Items (1), (2), (3), and (5) are the basic assumptions in fundamental mechanics, which can be easily calculated and have no significant impact on the results. Item (4) is proposed in the model of this study. The rectangular shape can be applied to small and medium-sized arch bridges with upper and lower bearings.

## 3. Analysis of Variable Cross-Section

In previous out-plane buckling analyses of the arch [33,34], the cross-sectional height $h$ is usually constant along the arch axis, and so the area $A$, the second moment of area $I$, the Saint-Venant torsional constant $J$, and the warping constant $I_w$ of the cross-section are also constant along the arch axis. In this paper, however, the cross-sectional height $h$ varies continuously along the arch axis, the relevant cross-sectional parameters also vary, and $h$ is defined by

$$h(\phi) = h_0 e^{a|\phi|} \tag{1}$$

with $a$ being the variable cross-sectional index, which can be given by

$$a = \frac{\ln(h_e/h_0)}{\Theta} \tag{2}$$

where $h_0$ and $h_e$ are the central and the end cross-sectional height, respectively. The average cross-sectional height hm can be obtained by integrating the cross-sectional height along the arch axis, as

$$h_m = \frac{\int_{-\Theta}^{\Theta} h(\phi)\mathrm{d}\phi}{\int_{-\Theta}^{\Theta} \mathrm{d}\phi} = \frac{h_0\left(e^{a\Theta} - 1\right)}{a\Theta} \tag{3}$$

Thus, the corresponding average cross-sectional radius of rotation $r_{xm}$ can be derived as

$$r_{xm} = \sqrt{\frac{1}{12}}h_m \tag{4}$$

For internal force analysis of variable cross-sectional arches, the calculation of the cross-sectional area $A$ and the second moment of area about its major principal axis $I_x$ are essential, and $A$ and $I_x$ are defined by

$$A(\phi) = \int_A \mathrm{d}A = bh_0 e^{a|\phi|} = A_0 e^{a|\phi|} \tag{5}$$

$$I_x(\phi) = \int_A y^2 \mathrm{d}A = \frac{bh_0^3 e^{3a|\phi|}}{12} = I_{x0} e^{3a|\phi|} \tag{6}$$

For out-plane buckling of variable cross-sectional arches having rectangular thin-walled sections, the effects of the cross-sectional second moment of area about its minor principal axis $I_y$, the cross-sectional warping constant $I_w$ and the cross-sectional Saint-Venant torsional constant $J$ should be considered in the total potential during out-plane buckling, which can be defined by

$$I_w(\phi) = \int_A [\omega(x,y)]^2 \mathrm{d}A = \frac{b^3 h_0^3 e^{3a|\phi|}}{144} = I_{w0} e^{3a|\phi|} \tag{7}$$

$$J(\phi) = \int_A \left[ \left(x - \frac{\partial \omega(x,y)}{\partial x}\right)^2 + \left(y + \frac{\partial \omega(x,y)}{\partial y}\right)^2 \right] \mathrm{d}A = \frac{bh_0 e^{a|\phi|}}{3} = J_0 e^{a|\phi|} \tag{8}$$

with the warping function being given by

$$\omega(x,y) = -xy \tag{9}$$

In addition, the shear correction coefficient of the rectangular thin-walled section is not affected by the variation of cross-sectional height, which is a constant and can be expressed as

$$\mu_y(\phi) = \frac{5}{6} \tag{10}$$

## 4. Out-Plane Buckling

### 4.1. Internal Force and Normal Stress of Variable Cross-Sectional Arch

Compared to the intricate process of finite element modeling, the analytical method only requires the dimensional information of the arch structure to calculate its critical load through formulas. Additionally, it facilitates optimization design analysis for structural computations. Therefore, in this paper, the analytical method is employed for out-plane buckling analysis of variable cross-section arches.

According to previous numerical analysis studies [28], a reasonable variation in the cross-sectional height of an arch can decrease its bending moment and thus increase its critical load of out-plane buckling. It is also known [29,30] that the accurate analytical solutions of internal forces and normal stress of a variable cross-sectional arch are indispensable for out-plane buckling analysis of the arch. These solutions can also be used to investigate the influences of the variable cross-section on the axial compression, bending moment, and normal stress, which in turn affect the critical buckling load of out-plane

buckling significantly. However, no analytical solutions of internal forces and normal stress of variable cross-sectional arches under a uniformly distributed radial local load are applicable in previous literature. The analytical solutions of internal forces for a variable cross-sectional arch can be obtained using Castigliano's second theorem.

The variable cross-sectional arch is a statically indeterminate structure. By cutting the crown of the variable cross-sectional arch into two parts, three redundant internal forces, including central axial compression $N_c$, central bending moments $M_c$, and central shear forces $Q_{yc}$ are created on the crown, which is plotted in Figure 2a. For the principle of structural symmetry, the central shear force $Q_{yc}$ should be equal to zero, so the mechanical model in Figure 2a can be simplified to that in Figure 2b. In addition, for equivalence of the cut arch to the original arch, the relative axial displacement $\Delta N_c$ and the relative rotation $\Delta M_c$ of the cut arch corresponding to central axial compression $N_c$ and central bending moments $M_c$ are also equal to zero, as

$$\Delta N_c = 0 \text{ and } \Delta M_c = 0, \tag{11}$$

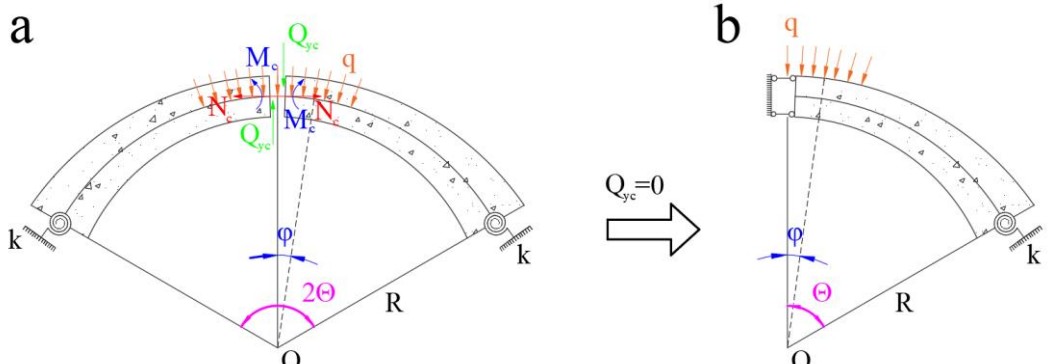

**Figure 2.** Basic system of a variable cross-sectional arch: (**a**) Before simplification; (**b**) After simplification.

According to Castigliano's second theorem, the relative axial displacement $\Delta N_c$ and the relative rotation $\Delta M_c$ can also be expressed as

$$\Delta N_c = \frac{\partial U_{ss0}}{\partial N_c} = 0 \text{ and } \Delta M_c = \frac{\partial U_{ss0}}{\partial M_c} = 0 \tag{12}$$

where $U_{ss0}$ is the pre-buckling total strain energy of the cut arch in Figure 2b, which is stated as

$$U_{ss0'} = \frac{1}{2} \int_0^\Theta \left[ \frac{N(\phi)^2}{EA(\phi)} + \frac{M(\phi)^2}{EI_x(\phi)} + \frac{\beta Q_y(\phi)^2}{EA(\phi)} \right] R d\phi + \frac{1}{2} \frac{M(\Theta)^2}{k(\Theta)} \tag{13}$$

Substituting the strain energy obtained from Equation (13) into Equation (12), the force method equilibrium equations can then be given by

$$\int_0^\Theta \left[ \frac{N(\phi)}{EA(\phi)} \frac{\partial N}{\partial M_c} + \frac{M(\phi)}{EI_x(\phi)} \frac{\partial M}{\partial M_c} + \frac{\beta Q_y(\phi)}{EA(\phi)} \frac{\partial Q_y}{\partial M_c} \right] R d\phi + \frac{M(\Theta)}{k(\Theta)} \frac{\partial M(\Theta)}{\partial M_c} = 0 \tag{14}$$

$$\int_0^\Theta \left[ \frac{N(\phi)}{EA(\phi)} \frac{\partial N}{\partial N_c} + \frac{M(\phi)}{EI_x(\phi)} \frac{\partial M}{\partial N_c} + \frac{\beta Q_y(\phi)}{EA(\phi)} \frac{\partial Q_y}{\partial N_c} \right] R d\phi + \frac{M(\Theta)}{k(\Theta)} \frac{\partial M(\Theta)}{\partial N_c} = 0 \tag{15}$$

By Equations (14) and (15), the linear homogeneous equations can be easily obtained, which can be expressed as

$$A_{11} M_c + A_{12} N_c + A_{13} = 0 \tag{16}$$

$$A_{21} M_c + A_{22} N_c + A_{23} = 0 \tag{17}$$

where the coefficients $A_{11}$, $A_{12}$, $A_{13}$, $A_{21}$, $A_{22}$, and $A_{23}$ are given in Appendix A. The central axial compression $N_c$ and the central bending moment $M_c$ can be obtained by solving the linear homogeneous Equations (16) and (17), as

$$N_c = \frac{A_{13}A_{21} - A_{11}A_{23}}{A_{11}A_{22} - A_{12}A_{21}} \tag{18}$$

$$M_c = \frac{A_{12}A_{23} - A_{13}A_{22}}{A_{11}A_{22} - A_{12}A_{21}} \tag{19}$$

Substituting two redundant internal forces $N_c$ and $M_c$ given by Equations (16) and (17), the axial compression $N$ and the bending moment $M$ of the whole variable cross-sectional arch can then be expressed as

$$N = \begin{cases} qR(E_2\cos\phi - \sin c\sin\phi) & \Theta \le \phi \le -c \\ qR(E_2\cos\phi - \cos c\cos\phi + 1) & -c \le \phi \le c \\ qR(E_2\cos\phi + \sin c\sin\phi) & c \le \phi \le \Theta \end{cases} \tag{20}$$

$$M = \begin{cases} qR^2(E_1 - E_2\cos\varphi + \sin c\sin\phi) & \Theta \le \phi \le -c \\ qR^2(E_1 - E_2\cos\varphi + \cos c\cos\phi - 1) & -c \le \phi \le c \\ qR^2(E_1 - E_2\cos\varphi - \sin c\sin\phi) & c \le \phi \le \Theta \end{cases} \tag{21}$$

where non-dimensional parameters $E_1$ and $E_2$ can be given by

$$E_1 = \frac{M_c + N_cR}{qR^2} \quad \text{and} \quad E_2 = \frac{N_c}{qR} \tag{22}$$

In addition, $\varepsilon_{ss0}$ is pre-buckling normal strain, which can be stated as

$$\begin{aligned} \varepsilon_{ss0}(\phi) &= \frac{dw(\phi)}{d\phi} - \frac{v(\phi)}{R} - \frac{y}{R}\frac{d\psi_y(\phi)}{d\phi} \\ &= -\frac{N(\phi)}{EA(\phi)} + \frac{yM(\phi)}{EI(\phi)} \end{aligned} \tag{23}$$

According to Equations (20), (21) and (23), stresses of cross-sectional upper and lower edge fibers for a rectangular variable cross-sectional arch, which can be derived, respectively, as

$$\sigma_{su}(\phi) = -\frac{N(\phi)}{A(\phi)} - \frac{h_0 e^{a|\phi|}M(\phi)}{2I(\phi)} \tag{24}$$

$$\sigma_{sl}(\phi) = -\frac{N(\phi)}{A(\phi)} + \frac{h_0 e^{a|\phi|}M(\phi)}{2I(\phi)} \tag{25}$$

*4.2. Critical Out-Plane Buckling Load of Variable Cross-Sectional Arch*

For an arch with rectangular thin-walled sections, where the cross-sectional height is much larger than the cross-sectional width, it is well known that when the internal forces induced by external loads in a variable cross-sectional arch reach a critical value, the arch may experience out-plane buckling. Therefore, the variation in cross-sectional height directly affects the internal forces within the arch, which in turn has a significant impact on the critical load for out-plane buckling. Additionally, based on previous research on the analysis of out-plane buckling in arches [12], the total potential energy of the variable cross-sectional arch during out-plane buckling can be expressed as

$$\begin{aligned} \Pi &= \int_{-\Theta}^{\Theta} \left\{ \frac{EI_y(\phi)}{2R}(\theta + \tilde{u}'')^2 + \frac{GJ(\phi)}{2R}(\theta' - \tilde{u}')^2 + \frac{EI_w(\phi)}{2R^3}(\theta'' - \tilde{u}'')^2 \right\} d\phi \\ &+ \int_{-\Theta}^{\Theta} \left\{ M(\phi)\left(\tilde{u}''\theta + \frac{\theta^2}{2} + \frac{\tilde{u}'^2}{2}\right) - N(\phi)R\left[\frac{\tilde{u}'^2}{2} + \frac{r_0^2(\phi)}{2R^2}(\theta' - \tilde{u}')^2\right] \right\} d\phi \end{aligned} \tag{26}$$

where

$$r_0(\phi) = \sqrt{\frac{I_x(\phi) + I_y(\phi)}{A(\phi)}} = \sqrt{\left(\frac{1}{12}h_0^2\right)e^{2a|\phi|} + \left(\frac{1}{12}b^2\right)} = \sqrt{r_{x0}^2 e^{2a|\phi|} + r_y^2} \tag{27}$$

with $r_{x0}$ is the in-plane gyration radius of the central cross-section and $r_y$ is out-plane gyration radius.

To determine the critical buckling load of out-plane buckling for variable cross-sectional arches subjected to a uniformly distributed radial local load, the required equilibrium differential equation can be derived by the Ritz method.

According to the above, the lateral displacement $\tilde{u}(\phi)$, and torsional angle $\theta(\phi)$ can be assumed, respectively, as

$$\tilde{u}(\phi) = \tilde{u}_1 \cos\frac{\pi\phi}{2\Theta} \tag{28}$$

$$\theta(\phi) = \theta_1 \cos\frac{\pi\phi}{2\Theta} \tag{29}$$

Substituting Equations (28) and (29) into the total potential energy of the variable cross-sectional arch during out-plane buckling obtained from Equation (26), Equation (30) can be rewritten as

$$\Pi = \frac{1}{2}\{\tilde{u}_1 \quad \theta_1\}(\boldsymbol{K_e} + qR\boldsymbol{K_g})\begin{Bmatrix} \tilde{u}_1 \\ \theta_1 \end{Bmatrix} \tag{30}$$

where the stiffness matrixes $\boldsymbol{K_e}$ and $\boldsymbol{K_g}$ can be expressed, respectively, as

$$\boldsymbol{K_e} = \begin{bmatrix} k_{e11} & k_{e12} \\ k_{e12} & k_{e22} \end{bmatrix} \tag{31}$$

$$\boldsymbol{K_g} = \begin{bmatrix} k_{g11} & k_{g12} \\ k_{g12} & k_{g22} \end{bmatrix} \tag{32}$$

with the elements of the stiffness matrixes $\boldsymbol{K_e}$ and $\boldsymbol{K_g}$ being given by

$$\begin{aligned} k_{e11} &= \frac{EI_{w0}\pi^4\left(\pi^2 e^{3a\Theta} - 18\Theta^2 a^2 - \pi^2\right)}{48R^3\theta\Theta^4 a(9\Theta^2 a^2 + \pi^2)} - \frac{EI_{y0}\pi^4\left(2\Theta^2 a^2 - e^{a\Theta}\pi^2 + \pi^2\right)}{16R\Theta^4 a(\Theta^2 a^2 + \pi^2)} \\ &+ \frac{GJ_0\pi^2\left(2e^{a\Theta}\Theta^2 a^2 + e^{a\Theta}\pi^2 - \pi^2\right)}{4R\Theta^2 a(\Theta^2 a^2 + \pi^2)} \end{aligned} \tag{33}$$

$$\begin{aligned} k_{e12} &= \frac{EI_{y0}\pi^2\left(2\Theta^2 a^2 - e^{a\Theta}\pi^2 + \pi^2\right)}{4R\Theta^2 a(\Theta^2 a^2 + \pi^2)} - \frac{EI_{w0}\pi^4\left(\pi^2 e^{3a\Theta} - 18\Theta^2 a^2 - \pi^2\right)}{48R^3\Theta^4 a(9\Theta^2 a^2 + \pi^2)} \\ &- \frac{GJ_0\pi^2\left(2e^{a\Theta}\Theta^2 a^2 + e^{a\Theta}\pi^2 - \pi^2\right)}{4R\Theta^2 a(\Theta^2 a^2 + \pi^2)} \end{aligned} \tag{34}$$

$$\begin{aligned} k_{e22} &= \frac{EI_y\pi^2\left(2\Theta^2 a^2 - e^{a\Theta}\pi^2 + \pi^2\right)}{4R\Theta^2 a(\Theta^2 a^2 + \pi^2)} - \frac{EI_w\pi^4\left(\pi^2 e^{3a\Theta} - 18\Theta^2 a^2 - \pi^2\right)}{48R^3\Theta^4 a(9\Theta^2 a^2 + \pi^2)} \\ &- \frac{GJ\pi^2\left(2e^{a\Theta}\Theta^2 a^2 + e^{a\Theta}\pi^2 - \pi^2\right)}{4R\Theta^2 a(\Theta^2 a^2 + \pi^2)} \end{aligned} \tag{35}$$

$$k_{g11} = \frac{\pi\left(2R^2 + r_y^2\right)}{4\Theta^2 R}\left[\pi(\pi^2 - 2\Theta^2)F_1 - F_2 - \pi c\right] + \frac{E_1\pi^2 R}{4\Theta} + \frac{\pi^2 r_{x0}^2\chi}{R} \tag{36}$$

$$\begin{aligned} k_{g12} &= \frac{\pi^2\left[\pi^2(R^2 + r_y^2) - 2\Theta^2 r_y^2\right]F_1 - \pi(R^2 - r_y^2)F_2}{4\Theta^2 R} + \frac{\pi^2 c(R^2 + r_0^2)}{4\Theta^2 R} \\ &- \frac{\pi^2 E_1 R}{4\Theta} - \frac{\pi^2 r_{x0}^2\chi}{R} \end{aligned} \tag{37}$$

and

$$k_{g22} = \frac{\pi^2 F_1 \left[\pi^2 r_0^2 + 2\Theta^2(2R^2 - r_y^2)\right]}{4\Theta^2 R} - \frac{c(\pi^2 r_y^2 + 4R^2\Theta^2)}{4R\Theta^2} + E_1\Theta R$$
$$+ \frac{F_2(4R^2\Theta^2 - \pi^2 r_y^2)}{4\Theta^2 R\pi} + \frac{\pi^2 r_{x0}^2 \chi}{R} \tag{38}$$

in which the parameters $F_1$, $F_2$, and $\chi$ are given by

$$F_1 = \frac{\sin c \cos\Theta - E_2 \sin\Theta}{\pi^2 - \Theta^2}, \quad F_2 = \frac{\Theta^3 \sin\left(\frac{c\pi}{\Theta}\right)}{\pi^2 - \Theta^2} \tag{39}$$

and

$$\chi = \frac{1}{2\Theta^2(4a^2+1)}\left[\frac{\pi^2 a(E_2 - \cos c)\left(4\Theta^2 a^2 + \pi^2 - 3\Theta^2\right)}{(4a^2+1)^2\Theta^4 + \pi^4 + 2\pi^2\Theta^2(4a^2-1)} - \frac{e^{2ca}}{4a}\right]$$
$$+ \frac{\Theta e^{2ca}}{2}\frac{2\Theta a\left[(4a^2+1)\Theta^2 - 3\pi^2\right]\cos\left(\frac{c\pi}{\Theta}\right) + \pi\left[(12a^2+1)\Theta^2 - \pi^2\right]\sin\left(\frac{c\pi}{\Theta}\right)}{3\Theta^6 a^2(4a^2+1)^2 + 2\pi^2\Theta^4(48a^4+1) + 4\pi^4\Theta^2(6a^2-1) + 2\pi^6}$$
$$- \frac{e^{2ca}(E_2 \sin\Theta - \cos\Theta \sin c)\left[2(4a^2+1)^2\Theta^4 + (4a^2-3)\pi^2\Theta^2 + \pi^4\right]}{4\Theta^2(4a^2+1)\left[\Theta^4(4a^2+1)^2 + 2\pi^2\Theta^2(4a^2-1) + \pi^4\right]}$$
$$- \frac{e^{2ca}a(E_2 \cos\Theta + \sin\Theta \sin c)\left[2\Theta^4(4a^2+1)^2 + \pi^2\Theta^2(12a^2-1) + \pi^4\right]}{2\Theta^2(4a^2+1)\left[\Theta^4(4a^2+1)^2 + 2\pi^2\Theta^2(4a^2-1) + \pi^4\right]}$$
$$+ \frac{\pi^2}{8\Theta^2 a(4\Theta^2 a^2 + \pi^2)} \tag{40}$$

In order to minimize the total potential energy of the variable cross-sectional arch during out-plane buckling, the principle of the Rayleigh–Ritz method can be used. The constraint conditions of the cross-sectional lateral displacement $\widetilde{u}(\phi)$ and torsional angle $\theta(\phi)$ of the arch are obtained as

$$\begin{cases} \frac{\partial\Pi(\widetilde{u}_1, \theta_1)}{\partial\widetilde{u}_1} = \left(k_{e11} + \frac{Q_{cr}}{N_y}k_{h11}\right)\widetilde{u}_1 + \left(k_{e12} + \frac{Q_{cr}}{N_y}k_{h12}\right)\theta_1 = 0 \\ \frac{\partial\Pi(\widetilde{u}_1, \theta_1)}{\partial\theta_1} = \left(k_{e12} + \frac{Q_{cr}}{N_y}k_{h12}\right)\widetilde{u}_1 + \left(k_{e22} + \frac{Q_{cr}}{N_y}k_{h22}\right)\theta_1 = 0 \end{cases} \tag{41}$$

where $Q_{cr}$ is the critical out-plane buckling load of the variable cross-sectional arch under a uniformly distributed radial local load, which can be expressed as

$$Q_{cr} = 2cRq_{cr} \tag{42}$$

$N_y$ is the critical buckling load of the simply supported column having the cross-sectional height $h_0$ and the length $S$, which can be given by

$$N_y = \frac{\pi^2 E I_{y0}}{S^2} \tag{43}$$

$k_{h11}$, $k_{h12}$, and $k_{h22}$ are the elements of the stiffness matrixes $\boldsymbol{K_h}$, and the stiffness matrixes $\boldsymbol{K_h}$ can be stated as

$$\boldsymbol{K_h} = \frac{N_y}{2c}\boldsymbol{K_g} \tag{44}$$

Equation (41) can be rewritten into a displacement vector-stiffness matrix form, as

$$\boldsymbol{K}\begin{Bmatrix} \widetilde{u}_1 \\ \theta_1 \end{Bmatrix} = \left(\boldsymbol{K_e} + \frac{Q_{cr}}{N_y}\boldsymbol{K_h}\right)\begin{Bmatrix} \widetilde{u}_1 \\ \theta_1 \end{Bmatrix} = 0 \tag{45}$$

Let the determinant of the stiffness matrix $K$ obtained from Equation (45) equal to zero, as

$$\left(k_{h11}k_{h22} - k_{h12}^2\right)\left(\frac{Q_{cr}}{N_y}\right)^2 + (k_{e11}k_{h22} - 2k_{e12}k_{h12} + k_{e22}k_{h11})\frac{Q_{cr}}{N_y} + k_{e11}k_{e22} - k_{e12}^2 = 0 \tag{46}$$

The non-dimensional critical buckling load $Q_{cr}/N_y$ can be obtained by solving the quadratic equation obtained from Equation (46).

## 5. Comparisons with Finite Element (FE) Results

### 5.1. Numerical Model of the Arch with Variable Cross-Section

According to the Timoshenko beam theory, the Beam188 element of ANSYS is suitable for constructing a variable cross-sectional arch model that considers shear deformation. Hence, the results obtained from ANSYS for variable cross-sectional arches with different cross-sectional indexes and included angles are compared against the analytical solution provided by Equation (46). This comparison aims to verify whether the non-dimensional critical out-plane buckling load $Q_{cr}/N_y$, obtained from Equation (46), can accurately predict the out-plane buckling behavior of variable cross-sectional arches subjected to a uniformly distributed radial local load.

In the variable cross-sectional arch model shown in Figure 3, the material's Young's modulus E is 34.5 GPa, the cross-sectional width $b$ is 0.15 m, the average cross-sectional height $h_m$ is 0.25 m, the in-plane slenderness ratio $S/r_{xm}$ is 50, and the out-plane slenderness ratio $S/r_y$ is 83.3. Additionally, the variable cross-sectional arch models have in-plane elastic rotation constraints and are subjected to a uniformly distributed radial local load. The flexibility coefficient of the elastic rotation constraint is $\zeta = 0.1$, and the ratio of the action length of the local load is $c/\Theta = 0.5$.

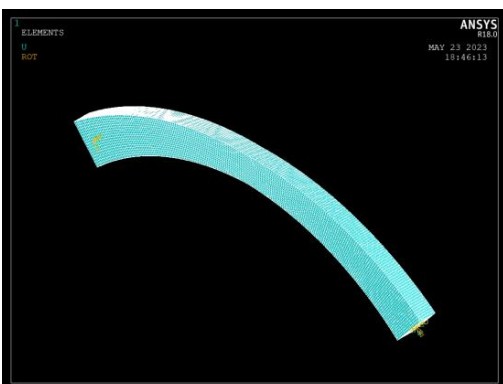

**Figure 3.** Variable cross-sectional arch model in ANSYS.

In addition, in order to more intuitively show the internal forces of the arch before buckling, the axial compression cloud image and bending moment cloud image of the arch structure with variable sections are shown in Figure 4a,b.

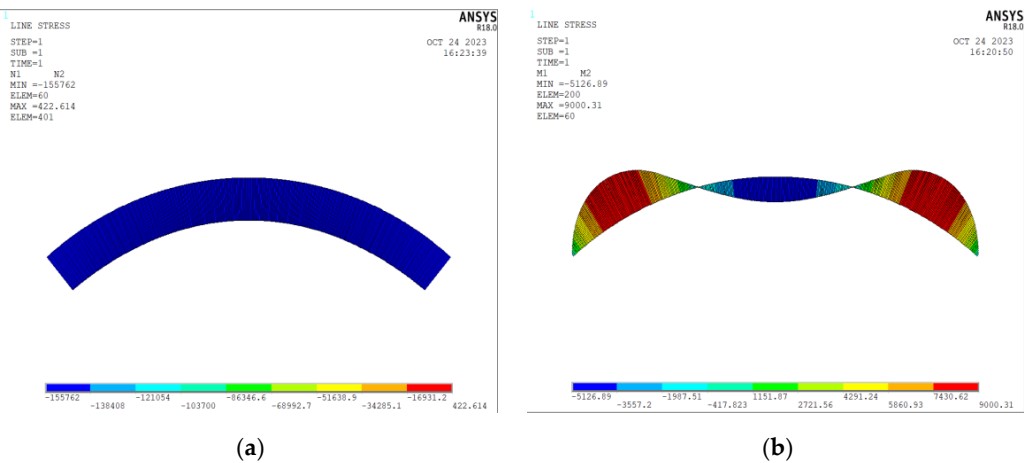

(**a**)　　　　　　　　　　　　　　　(**b**)

**Figure 4.** Internal force image of variable section arch: (**a**) axial compression cloud image; (**b**) bending moment cloud image.

At the same time, the stress condition of the arch before buckling plays a key role in the subsequent parametric analysis, so the stress cloud image is shown in Figure 5.

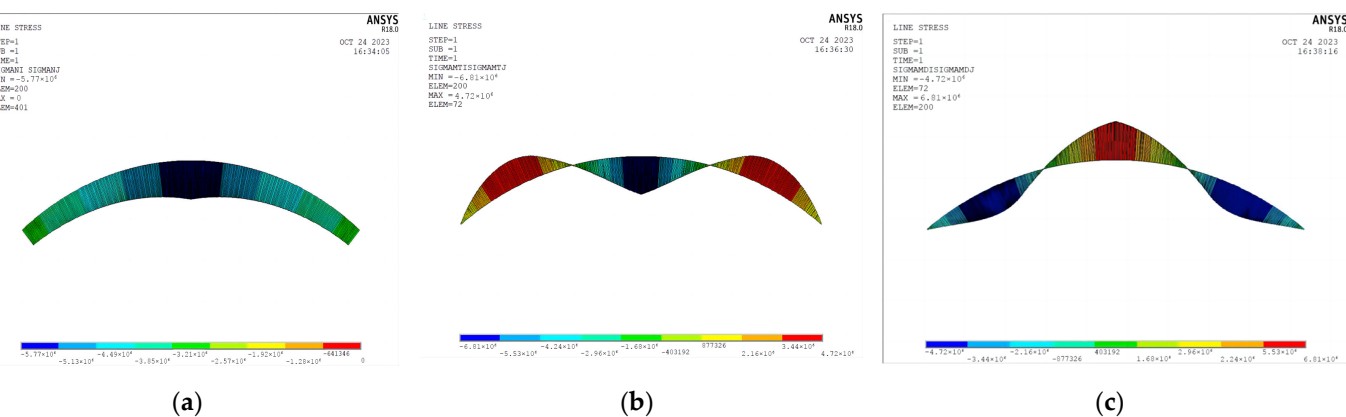

| (a) | (b) | (c) |

**Figure 5.** Stress image of a variable cross-section arch: (**a**) axial stress cloud image; (**b**) upper edge fibers stress cloud image; (**c**) lower edge fibers stress cloud image.

### 5.2. Comparative Analysis with Finite Element Results

Based on Equations (20) and (21), analytical solutions for internal forces before the arch buckles can be derived. The comparison with numerical solutions is shown in Figure 6. Analysis of Figure 6a indicates a good agreement between the analytical and numerical solutions for the axial force in the variable cross-section arch. Similarly, Figure 6b demonstrates a strong correspondence between the analytical and numerical solutions for the arch's bending moment. Therefore, the analytical solutions for internal forces in this study demonstrate a certain level of accuracy, providing a valuable reference for the design and research of arches.

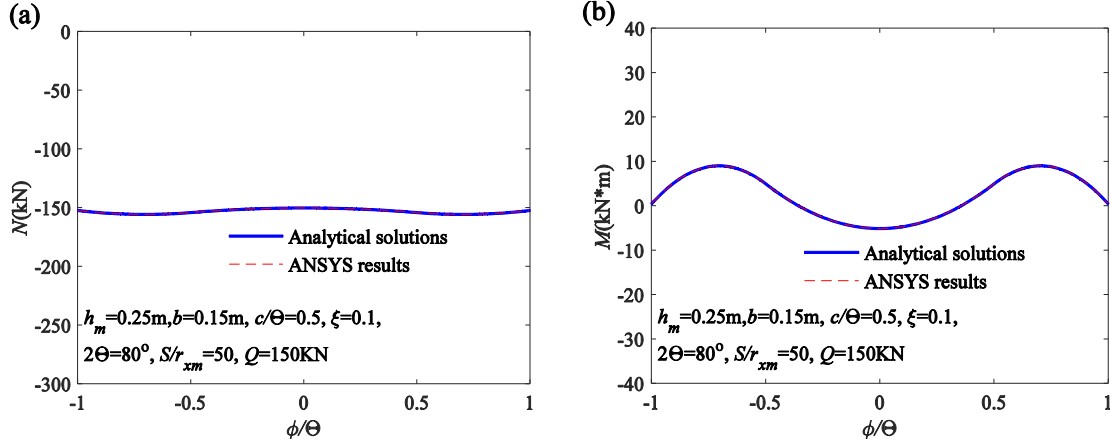

**Figure 6.** Comparison of internal force and finite element: (**a**) axial compression; (**b**) bending moment.

Based on Equations (23) to (25), analytical solutions for stress prior to the buckling of the variable cross-section arch can be obtained. The comparison with numerical solutions is illustrated in Figure 7. From Figure 7a, it can be observed that the axial strain in the variable cross-section arch aligns well with the numerical solution. Figure 7b demonstrates a relatively good agreement between the analytical solution for the upper edge fiber of the arch and the numerical solution. Likewise, Figure 7c indicates a favorable correspondence between the analytical solution for the lower edge fiber of the arch and the numerical solution. This affirms the accuracy of the strain analytical solutions presented in this study.

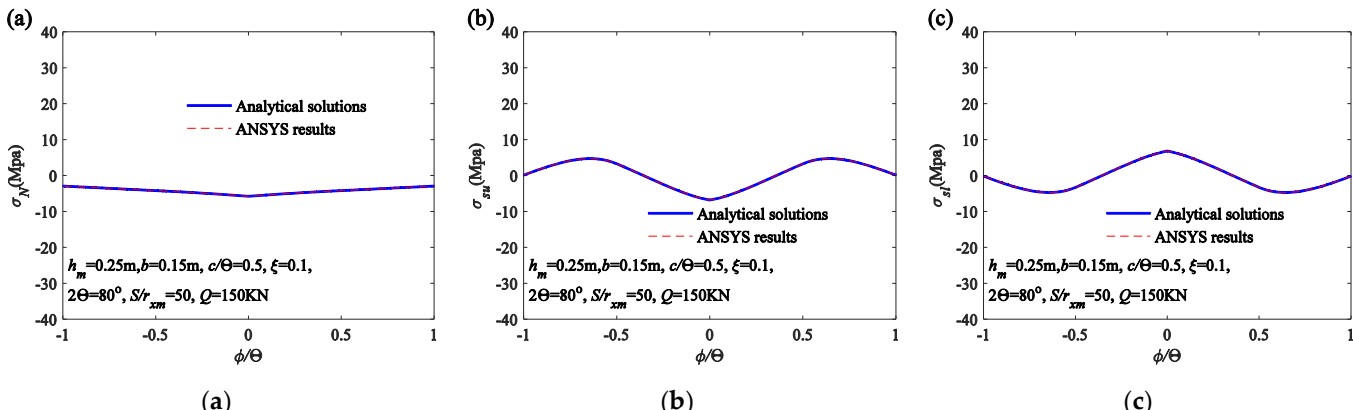

**Figure 7.** Comparison of stress and finite element: (**a**) axial stress $\sigma_N$; (**b**) upper-edge fibers $\sigma_{su}$; (**c**) lower edge fibers $\sigma_{sl}$.

The comparisons of internal forces of the variable cross-sectional arches between the analytical solutions obtained from Equations (20) and (21) and the ANSYS results are plotted in Figure 8a as the curves of the non-dimensional central axial compression $N_c/Q$ with included angle $2\Theta$, and in Figure 8b as the curves of the non-dimensional central bending moment $4 M_c/QL$ with included angle $2\Theta$ for variable cross-sectional arches having a different cross-sectional height ratio $h_e/h_0$, where $h_e$ and $h_0$ are the end and central cross-sectional height, respectively, and $Q = 2 qcR$.

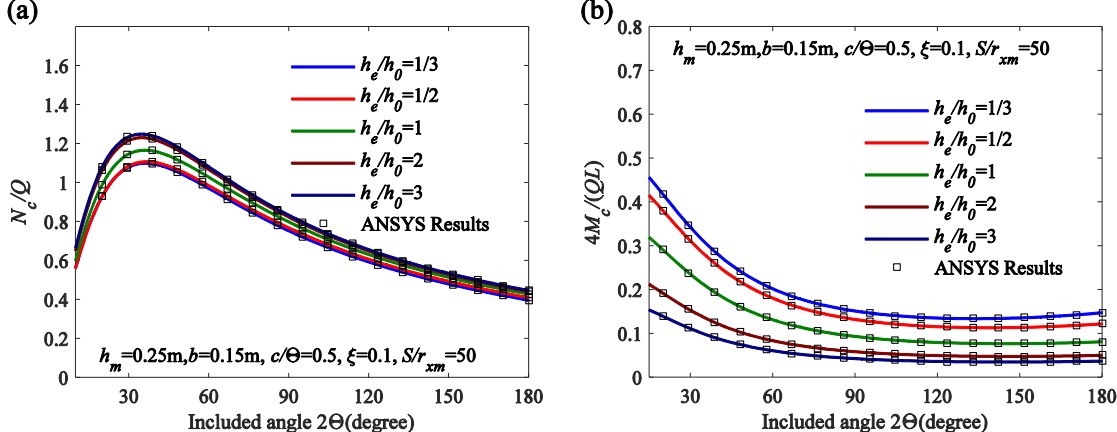

**Figure 8.** Analytical solution of internal force: (**a**) axial compression $N_c/Q$; (**b**) bending moment $4 M_c/(QL)$.

It can be seen from Figure 8a,b that the analytical solutions of the axial compressions and bending moments obtained from Equations (20) and (21) agree excellently against the ANSYS results, so the solutions obtained from Equations (20) and (21) can predict the internal forces of variable cross-sectional arches under a uniformly distributed radial local load.

In addition, it can be indicated from Figure 8a that variations of axial compression of variable cross-sectional arches with the included angle are similar to those of uniform cross-sectional arches. When the included angle $2\Theta$ increases, the non-dimensional central axial compression increases significantly at first and then slightly decreases. However, for the same volume of variable cross-sectional arches and uniform cross-sectional arches, when the cross-sectional height ratio $h_e/h_0$ increases, the non-dimensional central axial compression increases. It can be indicated from Figure 4b that variations of bending moment of variable cross-sectional arches with the included angle are similar to those of uniform cross-sectional arches. When the included angle $2\Theta$ increases, the non-dimensional central bending

moment decreases significantly at first and then slightly increases. However, for the same volume of variable cross-sectional arches and uniform cross-sectional arches, when the cross-sectional height ratio $h_e/h_0$ increases, the non-dimensional central bending decreases.

The comparisons of critical out-plane buckling loads of the variable cross-sectional arches between the analytical solutions obtained from Equation (46) and the ANSYS results are plotted in Figure 9 as the curves of the non-dimensional critical out-plane buckling load $Q_{cr}/N_y$ with included angle $2\Theta$ for variable cross-sectional arches having a different cross-sectional height ratio $h_e/h_0$.

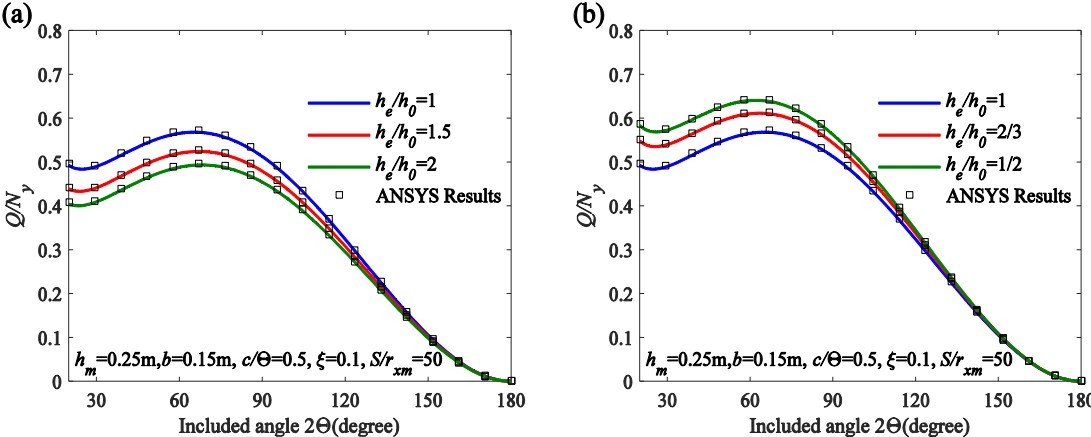

**Figure 9.** Analytical solution and finite element results of vault: (**a**) included angle; (**b**) flexibility parameter.

It can be seen from Figure 9 that the analytical solutions of the critical out-plane buckling load obtained from Equation (46) agree excellently against the ANSYS results. Therefore, the solutions obtained from Equation (46) can predict the critical out-plane buckling loads of variable cross-sectional arches under a uniformly distributed radial local load.

In addition, for the same volume of variable cross-sectional arches and uniform cross-sectional arches, Figure 9 shows that when the cross-sectional height ratio $h_e/h_0$ increases, the non-dimensional critical out-plane buckling load increases.

## 6. Parametric Analysis

### 6.1. Parametric Analysis of Stresses and Internal Forces

The pre-buckling axial compression and bending moment behavior and effects of cross-sectional height ratio $h_e/h_0$, in-plane slenderness ratio $S/r_{xm}$, flexibility coefficient of elastic rotation constraint $\zeta$, and the ratio of action length of local load on the internal forces of variable cross-sectional arches are based on the analytical solutions obtained from Equations (20) and (21), while the pre-buckling stresses behavior of variable cross-sectional arches is based on the analytical solutions obtained from Equations (24) and (25).

To demonstrate the influence of the cross-sectional height ratio $h_e/h_0$ on pre-buckling stresses, distributions of stresses for variable cross-sectional arches with different cross-sectional height ratios (i.e., $h_e/h_0 = 0.5$, 1, and 2) are plotted in Figure 10a,b. The figures depict distributions of the stresses in the cross-sectional lower and upper edge fibers (i.e., $\sigma_{sl}$ and $\sigma_{su}$) against the non-dimensional angular coordinates of the cross-section $\phi/\Theta$ for shallow variable cross-sectional arches with an included angle of $2\Theta = 40°$, respectively. Similarly, distributions of $\sigma_{sl}$ and $\sigma_{su}$ against $\phi/\Theta$ are shown in Figure 10c,d for deep variable cross-sectional arches with an included angle of $2\Theta = 120°$.

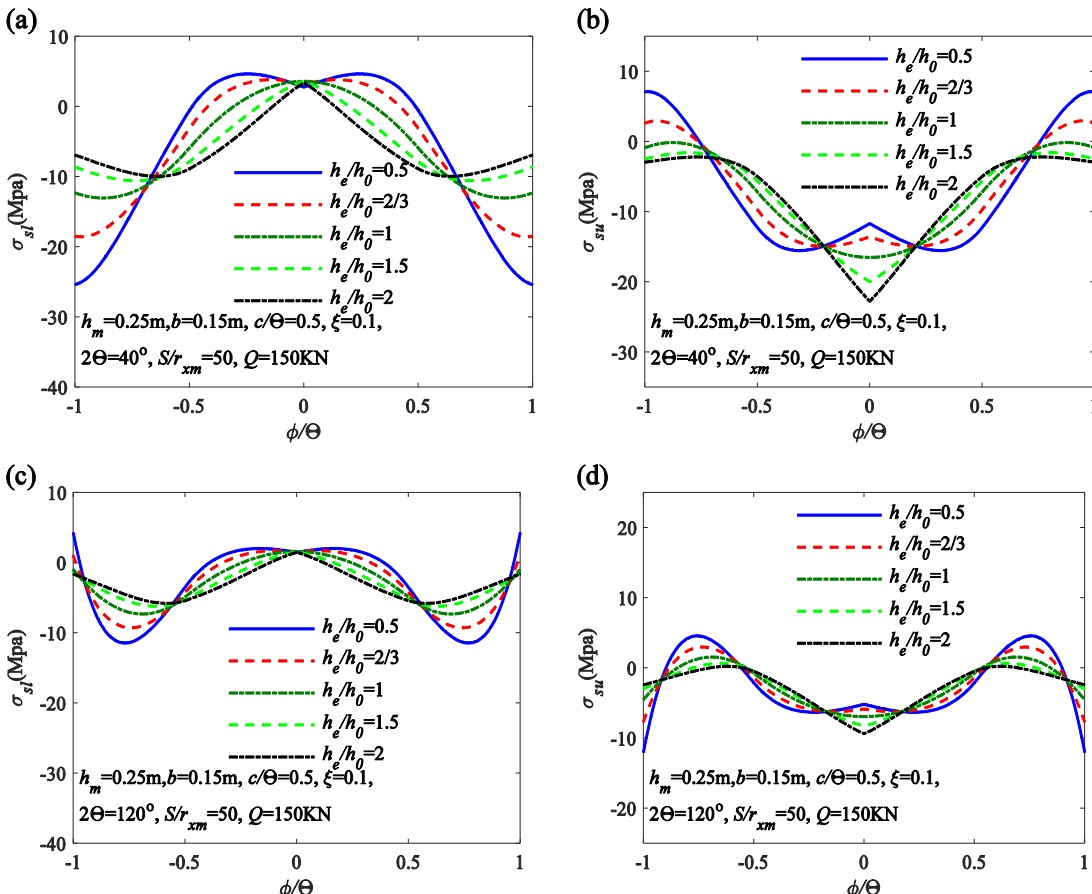

**Figure 10.** Distributions of the stresses in the cross-sectional lower edge fibers $\sigma_{sl}$ and upper edge fibers $\sigma_{su}$ along arch length: (**a**) lower edge fibers $\sigma_{sl}$ for shallow arch, $2\Theta = 40°$; (**b**) upper edge fibers $\sigma_{su}$ for shallow arch, $2\Theta = 40°$; (**c**) lower edge fibers $\sigma_{sl}$ for deep arch, $2\Theta = 120°$; (**d**) upper edge fibers $\sigma_{su}$ for deep arch, $2\Theta = 120°$.

Figure 10 demonstrates that as the cross-sectional height ratio of the variable cross-sectional arch increases, the stress distribution in the arch becomes more uniform. In addition, for shallow variable cross-sectional arches, Figure 10a,b indicated that the stresses at the upper and lower edges of the cross-section are primarily compressive. However, there may be instances of tensile stresses present at the upper edge of the cross-section in the end support segments, as well as at the lower edge of the cross-section in the crown segments. As the cross-sectional height ratio of the variable cross-sectional arch increases, the segment of the arch axis experiencing tensile stresses gradually decreases. Therefore, designing variable cross-sectional arches with a larger cross-sectional height ratio is advantageous for structures with lower tensile strength, such as reinforced concrete structures. For deep variable cross-sectional arches, Figure 10c,d indicated that the upper and lower edges of the cross-section also primarily experience compressive stresses. However, apart from the potential occurrence of tensile stresses at the lower edge of the cross-section in the end support segments and at the lower edge of the cross-section in the crown segments, there might also be instances of tensile stresses at the upper edge of the cross-section near the end support segments. This phenomenon arises from the presence of inflection points in the end support segments of the arch. As the cross-sectional height ratio $h_e/h_0$ of the variable cross-sectional arch increases, the segments of the arch axis experiencing tensile stresses also gradually decrease.

To demonstrate the effects of cross-sectional height ratio $h_e/h_0$ and in-plane slenderness ratio $S/r_{xm}$ on the pre-buckling internal force, the analytical solutions of internal forces of variable cross-sectional arches having different cross-sectional height ratios (i.e.,

$h_e/h_0 = 0.5$, 1 and 2) are plotted in Figure 11a,b,c, and d as follows: In Figure 11a, as the curves of the non-dimensional central axial compression $N_c/Q$ versus the in-plane slenderness ratio $S/r_{xm}$ for shallow variable cross-sectional arches having a included angle $2\Theta = 40°$. In Figure 11b, as the curves of the non-dimensional central bending moment $4M_c/QL$ versus the in-plane slenderness ratio $S/r_{xm}$ for shallow variable cross-sectional arches having a included angle $2\Theta = 40°$. In Figure 11c, as the curves of $N_c/Q$ versus $S/r_{xm}$ for deep variable cross-sectional arches having a included angle $2\Theta = 120°$. In Figure 10d, as the curves of $4M_c/QL$ versus $S/r_{xm}$ for deep variable cross-sectional arches having a included angle $2\Theta = 120°$.

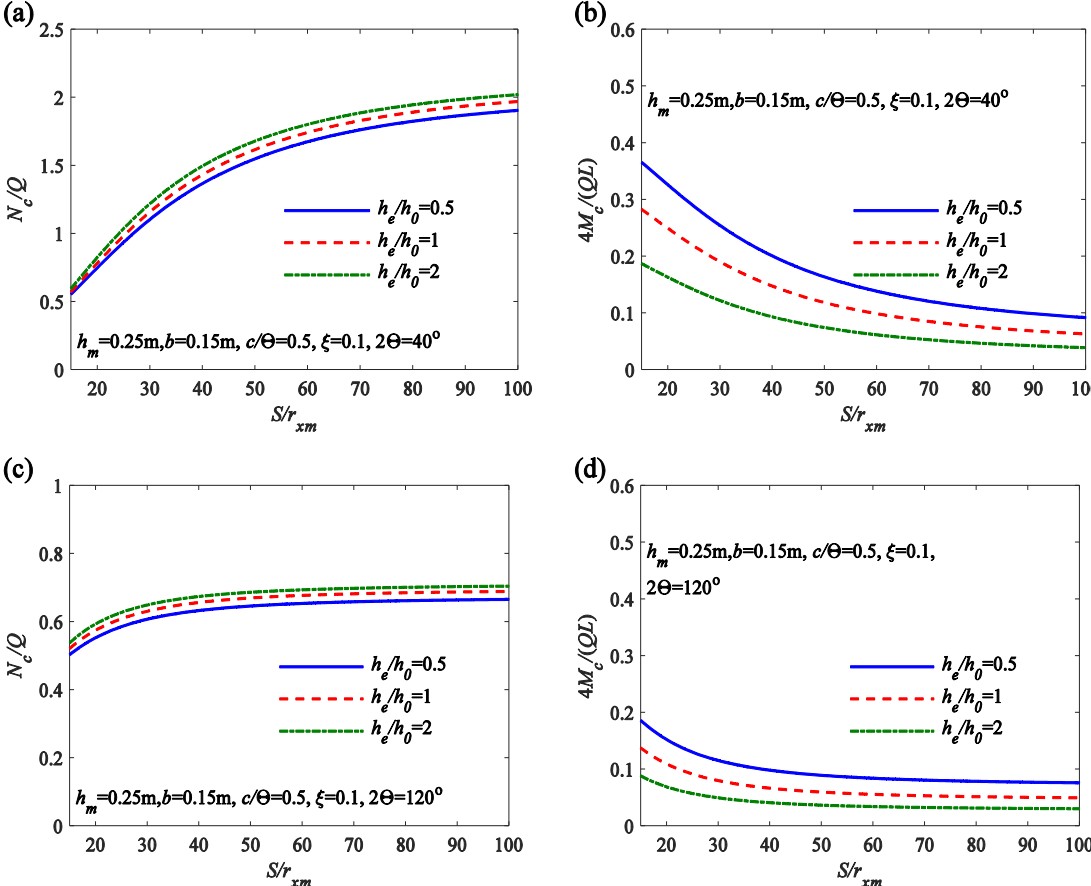

**Figure 11.** Influence of slenderness ratio S/rxm on the forces in the arch with variable cross-section: (**a**) axial compression for $2\Theta = 40°$; (**b**) bending moment for $2\Theta = 40°$; (**c**) axial compression for $2\Theta = 120°$; (**d**) bending moment for $2\Theta = 120°$.

Figure 11a,b demonstrate the variations in internal forces for shallow variable cross-sectional arches under the slenderness ratio $S/r_{xm}$. Figure 11a shows that as the slenderness ratio $S/r_{xm}$ increases, the axial compression $N_c/Q$ in the shallow variable cross-sectional arch also increases. Figure 11b indicates that as the slenderness ratio $S/r_{xm}$ increases, the bending moment $4M_c/QL$ in the shallow arch gradually decreases. Additionally, as the cross-sectional height ratio $h_e/h_0$ of the variable cross-sectional arch increases, larger axial compressions $N_c/Q$ and smaller bending moments $4M_c/QL$ can be achieved, contributing to the compressive performance of the arch structure. Figure 11c,d demonstrate the variations in internal forces for deep variable cross-sectional arches under the slenderness ratio $S/r_{xm}$. Both Figure 11c,d lead to similar conclusions as shallow variable cross-sectional arches, where an increase in the slenderness ratio $S/r_{xm}$ corresponds to an increase in axial compression $N_c/Q$ and a decrease in the bending moment $4M_c/QL$ for deep variable cross-sectional arches. Similarly, with an increase in the cross-sectional height ratio $h_e/h_0$ of the variable cross-sectional arch, greater axial compressions and smaller bending moments

4 $M_c/QL$ can be obtained for deep variable cross-sectional arches as well. Furthermore, it is noticeable that shallow variable cross-sectional arches exhibit more pronounced changes in internal forces than those for deep variable cross-sectional arches in response to variations in the cross-sectional height ratio $h_e/h_0$ of the variable cross-sectional arch.

To explore the distribution law of internal force along the arch length, the distribution of non-dimensional axial compression $N/Q$ and non-dimensional central bending moment $4 M_c/(QL)$ along the length of variable section arch under a localized uniform radial load is shown in Figure 12. Figure 12a–d have the same parameters, mainly including the average cross-sectional height $h_m = 0.25$ m, cross-section width $b = 0.15$ m, the ratio of the action length $c/\Theta = 0.5$, the end rotation constraint $\zeta = 0.1$ and the slenderness ratio $S/r_{xm} = 50$. Meanwhile, the included angle of Figure 12a,b is 40°, and the included angle of Figure 12c,d is 120°.

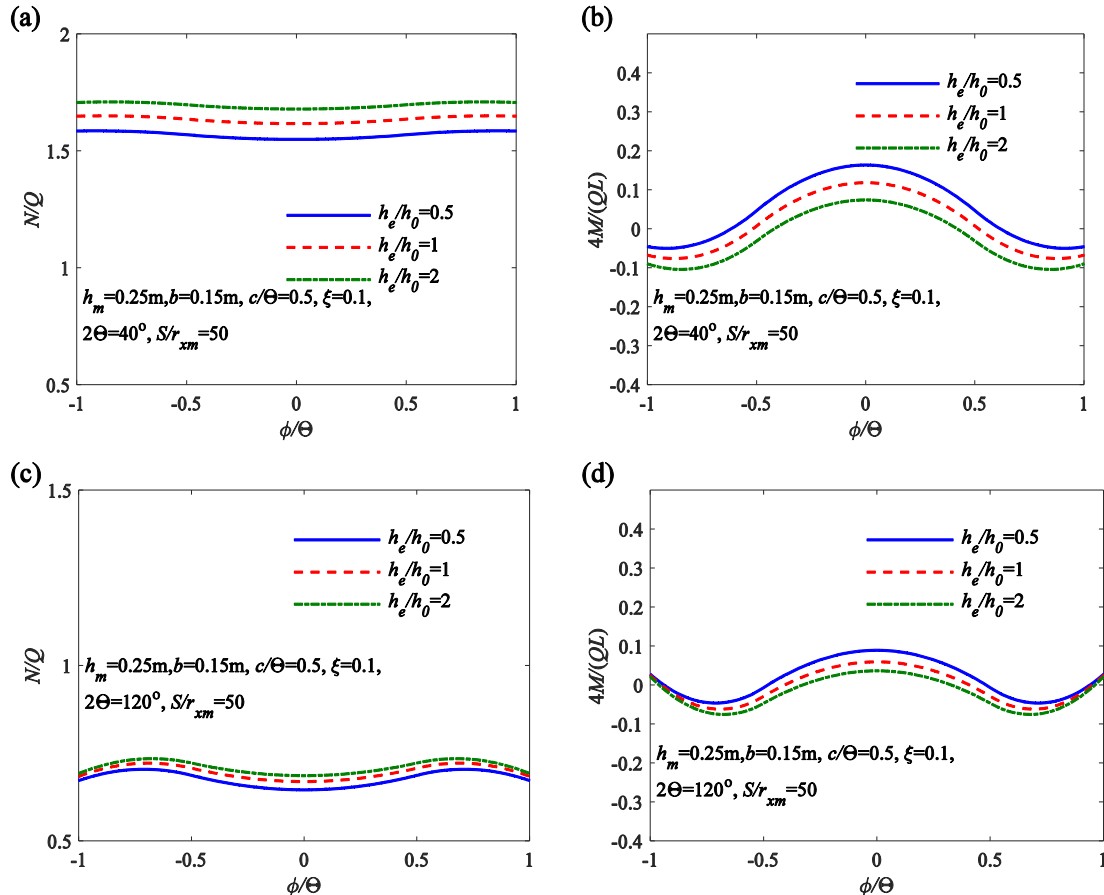

**Figure 12.** Distributions non-dimensional axial compression of $N/QL$ and non-dimensional bending moment of 4 $M/QL$ along arch length: (**a**) Axial compression for $2\Theta = 40°$; (**b**) Bending moment for $2\Theta = 40°$; (**c**) Axial compression for $2\Theta = 120°$; (**d**) Bending moment $2\Theta = 120°$.

It can be seen from Figure 12 that for a variable cross-sectional arch, the non-dimensional axial compression $N/Q$ and non-dimensional central bending moment $4 M_c/(QL)$ along the arch length is nonuniform. From Figure 12a,b, it can be observed that for shallow variable cross-sectional arches with $2\Theta = 40°$, the axial compression $N/Q$ gradually increases as the cross-sectional height ratio $h_e/h_0$ of the variable cross-sectional arch increases, while the bending moment $4 M_c/(QL)$ decreases. Similarly, Figure 8c,d show that for deep variable cross-sectional arches with $2\Theta = 120°$, the variations in axial compression $N/Q$ and bending moment $4 M_c/(QL)$ follow the same trend as in shallow variable cross-sectional arches, namely, an increase in the cross-sectional height ratio $h_e/h_0$ leads to an increase in axial compression $N/Q$ and a decrease in bending moment $4 M_c/(QL)$. Additionally, it should be noted that the effects of the cross-sectional height ratio $h_e/h_0$ on internal forces

of shallow variable cross-sectional arches are more significant than those effects on deep variable cross-sectional arches.

To demonstrate the effects of the cross-sectional height ratio $h_e/h_0$ and the localized parameter $c/\Theta$ on the pre-buckling internal force, the analytical solutions of internal forces of variable cross-sectional arches having different cross-sectional height ratios (i.e., $h_e/h_0 = 0.5$, 1 and 2) are plotted in Figure 13a,b,c, and d as follows: In Figure 13a, as the curves of the non-dimensional central axial compression $N_c/Q$ versus the localized parameter $c/\Theta$ for shallow variable cross-sectional arches having a included angle $2\Theta = 40°$. In Figure 13b, as the curves of the non-dimensional central bending moment $4M_c/QL$ versus the localized parameter $c/\Theta$ for shallow variable cross-sectional arches having a included angle $2\Theta = 40°$. In Figure 13c, as the curves of $N_c/Q$ versus $c/\Theta$ for deep variable cross-sectional arches having a included angle $2\Theta = 120°$. In Figure 13d, as the curves of $4M_c/QL$ versus $c/\Theta$ for deep variable cross-sectional arches having a included angle $2\Theta = 130°$.

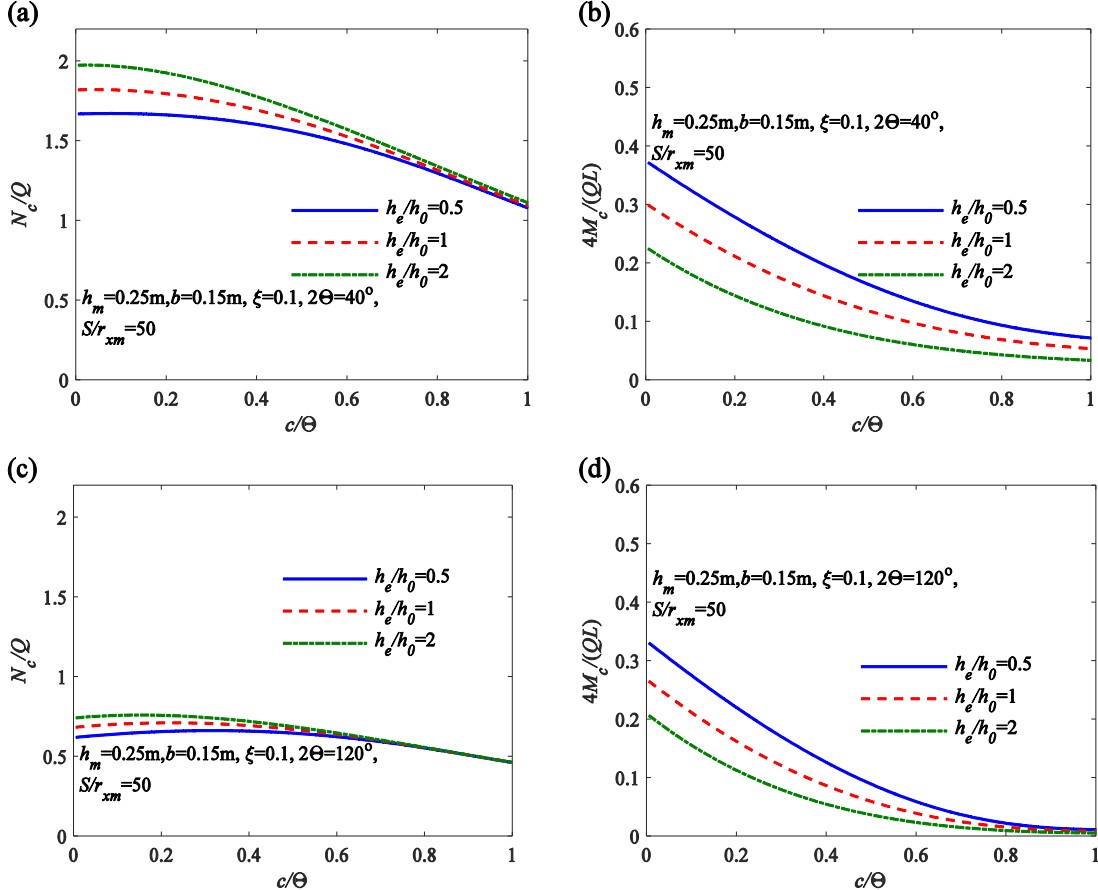

**Figure 13.** Influence of the localized parameter $c/\Theta$ on the forces in the arch with variable cross-section: (**a**) axial compression for $2\Theta = 40°$; (**b**) bending moment for $2\Theta = 40°$; (**c**) axial compression for $2\Theta = 120°$; (**d**) bending moment for $2\Theta = 120°$.

Figure 13a illustrates that as the ratio of the action length $c/\Theta$ increases, the axial compression of the arch with different cross-sectional height ratios $h_e/h_0$ gradually decreases, while the axial compression increases as the cross-sectional height ratio of the variable cross-sectional arch increases. In Figure 13c, it can be observed that for the deep arch with $2\Theta = 120°$, the change in the ratio of the action length $c/\Theta$ has little impact on the axial compression of the arch with different cross-sectional height ratios $h_e/h_0$. When the ratio of the action length $c/\Theta$ exceeds 0.6, the axial compression $N/Q$ is nearly unaffected by the cross-sectional height ratio $h_e/h_0$. Moreover, Figure 13b,d reveal that as the ratio of the action length $c/\Theta$ increases, the bending moment of the arch with different cross-

sectional height ratios $h_e/h_0$ gradually decreases, and the bending moment increases as the cross-sectional height ratio $h_e/h_0$ of the variable cross-sectional arch decreases.

To explore the effects of cross-sectional height ratio $h_e/h_0$ and the non-dimensional flexibility of the flexibility coefficient of elastic rotation constraint $\zeta$ on the pre-buckling internal force, the analytical solutions of internal forces of variable cross-sectional arches having different cross-sectional height ratios (i.e., $h_e/h_0 = 0.5$, 1 and 2) are plotted in Figure 14a,b,c, and d as follows: In Figure 14a, as the curves of the non-dimensional central axial compression $N_c/Q$ versus $\zeta$ for shallow variable cross-sectional arches having a included angle $2\Theta = 40°$. In Figure 14b, as the curves of the non-dimensional central bending moment $4\,M_c/QL$ versus $\zeta$ for shallow variable cross-sectional arches having a included angle $2\Theta = 40°$. In Figure 14c, as the curves of $N_c/Q$ versus $\zeta$ for deep variable cross-sectional arches having a included angle $2\Theta = 120°$. In Figure 14d, as the curves of 4 $M_c/QL$ versus $\zeta$ for deep variable cross-sectional arches having a included angle $2\Theta = 120°$.

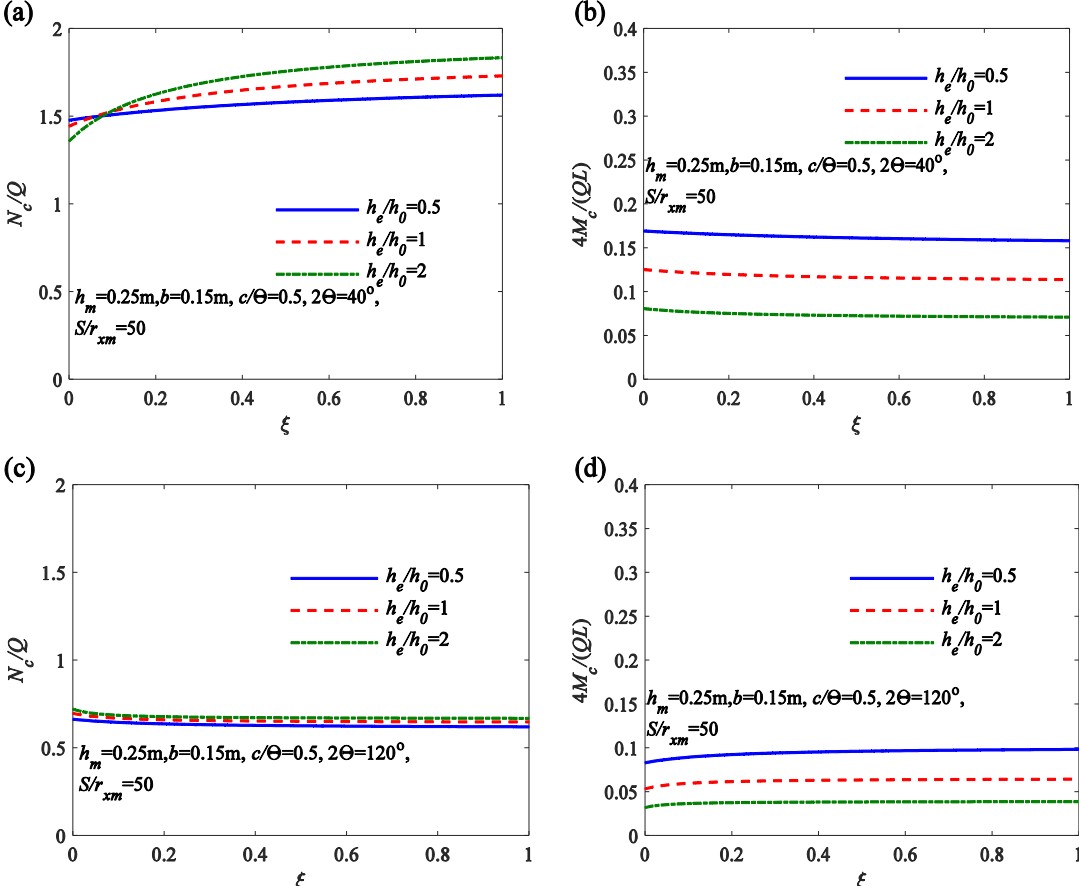

**Figure 14.** Influence of the non-dimensional flexibility of the flexibility coefficient of elastic rotation constraint $\zeta$ on the internal forces: (**a**) axial compression for $2\Theta = 40°$; (**b**) bending moment for $2\Theta = 40°$; (**c**) axial compression for $2\Theta = 120°$; (**d**) bending moment for $2\Theta = 120°$.

Figure 14a,b illustrate the variations of axial compression $N_c/Q$ and bending moment 4 $M_c/QL$ for shallow variable cross-sectional arches with different cross-sectional height ratios $h_e/h_0$ as the flexibility coefficient of elastic rotation constraint $\zeta$ change. According to Figure 14a, it can be observed that as the cross-sectional height ratio $he/h_0$ increases, the axial compression $N_c/Q$ of the shallow variable cross-sectional arch gradually increases. Furthermore, the axial compression $N_c/Q$ of the arch also increases with an increase in the flexibility coefficient of elastic rotation constraint $\zeta$, and the rate of increase is higher for arches with larger cross-sectional height ratios $h_e/h_0$. From Figure 14b, it can be seen that as the cross-sectional height ratio $h_e/h_0$ increases, the bending moment 4 $M_c/QL$ of the shallow variable cross-sectional arch slightly decreases. The bending moment 4 $M_c/QL$ of the arch

decreases with an increase in the flexibility coefficient of elastic rotation constraint $\zeta$, but the decrease is not significant. Figure 14c,d represent the variations in axial compression $N_c/Q$ and bending moment $4M_c/QL$ of deep variable cross-sectional arches with different cross-sectional height ratios $h_e/h_0$ under changing flexibility coefficient of elastic rotation constraint $\zeta$. From Figure 14c, it can be observed that as the cross-sectional height ratio $h_e/h_0$ increases, the axial compression $N_c/Q$ of deep variable cross-sectional arches slightly increases. Additionally, the axial compression decreases slightly with an increase in the end rotational restraint $\zeta$. Figure 14d indicates that as the cross-sectional height ratio $h_e/h_0$ increases, the bending moment $4M_c/QL$ of shallow variable cross-sectional arches decreases to some extent. Moreover, with an increase in the end rotational restraint $\zeta$, the bending moment $4M_c/QL$ of the arch increases, but the change is not significant.

According to Figures 11–14, it can be seen that compared with an arch with a uniform cross-section height ratio of $h_e/h_0 = 1$, for a variable cross-section arch with $h_e/h_0$ greater than 1, the axial compression $N_c/Q$ of the arch is larger and the bending moment $4M_c/QL$ is smaller. However, for a variable cross-section arch with $h_e/h_0$ less than 1, the axial compression $N_c/Q$ is smaller, and the bending moment $4M_c/QL$ is larger. Therefore, the design of an arch with $h_e/h_0$ greater than 1 is more reasonable, with more uniform internal forces and better mechanical performance.

### 6.2. Parametric Analysis of Critical Buckling Load

Typical variations of the critical out-plane buckling load $Q_{cr}/N_y$ with the slenderness ratio $S/r_{xm}$ obtained from Equation (46) are plotted in Figure 15a for shallow variable cross-sectional arches (included angle $2\Theta = 40°$) and in Figure 15b for deep variable cross-sectional arches (included angle $2\Theta = 120°$), where the localized parameter $c/\Theta = 0.5$, and the non-dimensional flexibility of the flexibility coefficient of elastic rotation constraint $\zeta = 0.1$.

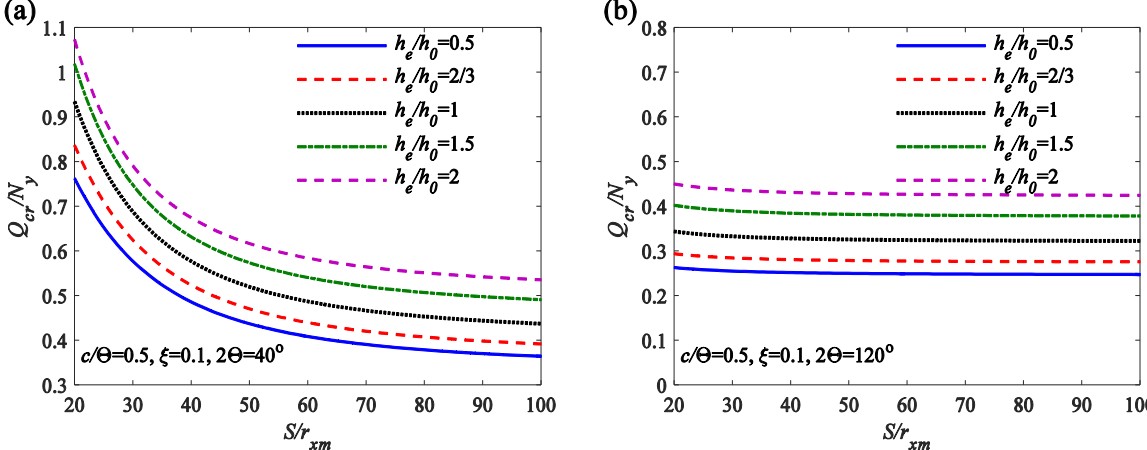

**Figure 15.** Influence of slenderness ratio $S/r_{xm}$ on critical out-plane buckling load: (**a**) $2\Theta = 40°$; (**b**) $2\Theta = 120°$.

It can be observed from Figure 15a that for shallow variable cross-sectional arches, the critical out-plane buckling load $Q_{cr}/N_y$ decreases as the slenderness ratio $S/r_{xm}$ increases. On the other hand, Figure 15b shows that for deep variable cross-sectional arches, the non-dimensional out-plane buckling load $Q_{cr}/N_y$ is nearly unaffected by the slenderness ratio $S/r_{xm}$. Additionally, a larger cross-sectional height ratio $he/h_0$ leads to a higher out-plane buckling load, indicating a greater structural stability.

Figure 16 offers insight into the effects of the flexibility coefficient of elastic rotation constraint $\zeta$ on critical out-plane buckling load $Q_{cr}/N_y$, where the localized parameter $c/\Theta = 0.5$ and the slenderness ratio $S/r_{xm} = 50$. Figure 16a reveals that the non-dimensional out-plane buckling load of a shallow variable cross-sectional arch (included angle $2\Theta = 40°$) is affected by the flexibility coefficient of elastic rotation constraint $\zeta$, and it can be concluded

from Figure 16a that the critical out-plane buckling load $Q_{cr}/N_y$ decreases with the increase of the flexibility coefficient of elastic rotation constraint $\zeta$, which is fast in the beginning and slow in the later period. However, the critical out-plane buckling load $Q_{cr}/N_y$ of a deep variable cross-sectional arch (included angle $2\Theta = 120°$) has little effect on the variation of the flexibility coefficient of elastic rotation constraint $\zeta$ as shown in Figure 16b. In addition, consistent with Figure 15, a larger cross-sectional height ratio $h_e/h_0$ can result in a greater critical out-plane buckling load $Q_{cr}/N_y$.

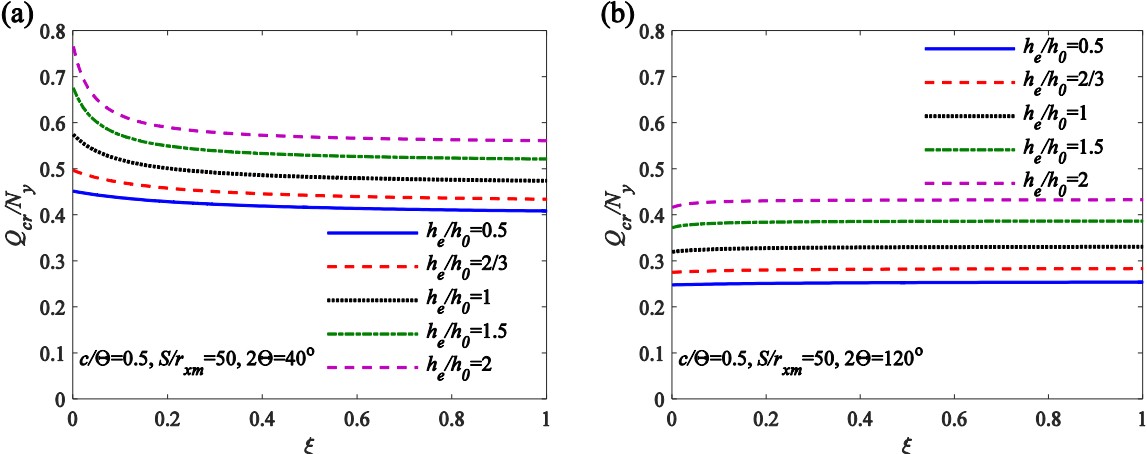

**Figure 16.** Influence of the flexibility coefficient of elastic rotation constraint $\zeta$ on critical out-plane buckling load: (**a**) $2\Theta = 40°$; (**b**) $2\Theta = 120°$.

## 7. Conclusions

This paper presented analytical investigations on the elastic out-plane buckling of circular variable cross-sectional arches. By considering the variable cross-sectional form using the natural constant $e$, analytical solutions for the pre-buckling stresses, internal forces, and critical out-plane buckling load of the arches were obtained. The analytical solutions for pre-buckling internal forces and critical out-plane buckling load were compared with the ANSYS results, and a high level of agreement was observed between them. This demonstrates that the analytical solutions presented in this paper were capable of accurately predicting the elastic out-plane buckling behavior. In addition, by conducting parameter analysis of pre-buckling stresses, internal forces, and critical out-plane buckling load, the following was found:

(1)  As the cross-sectional height ratio of the variable cross-sectional arch increases, the segments of the arch axis experiencing compressive stress increase, while the segments experiencing tensile stress gradually decrease. These result in a more uniform distribution of stress in the arch.

(2)  Through the analysis of the effects of the in-plane slenderness ratio $S/r_{xm}$, the localized parameter $c/\Theta$, the flexibility coefficient of the flexibility coefficient of elastic rotation constraint $\zeta$, and the variation of internal forces along the arch axis, it can be observed that compared to arches with a uniform cross-sectional height ratio $h_e/h_0 = 1$, variable cross-sectional arches with $h_e/h_0$ greater than 1 exhibit larger axial compression $N_c/Q$ and smaller bending moment $4\,M_c/(QL)$. However, for variable cross-sectional arches with $he/h_0$ less than 1, the axial compression $N_c/Q$ is smaller, and the bending moment $4\,M_c/(QL)$ is larger. Therefore, properly designed variable cross-sectional arches exhibit more uniform internal force distribution and better pre-buckling performance.

(3)  Through the parametric analysis of the critical out-plane buckling load, it has been found that compared to arches with a uniform cross-section height ratio of $h_e/h_0 = 1$, variable cross-section arches with a cross-section height ratio $h_e/h_0$ greater than 1 can achieve a larger critical out-plane buckling load, indicating a higher level of stability. These findings provide valuable insights for structural design.

**Author Contributions:** Conceptualization, A.J., D.D., W.D., X.Y. and H.L.; methodology, A.J., D.D., W.D., X.Y. and H.L.; software, A.J., D.D., W.D., X.Y. and H.L.; formal analysis, A.J. and H.L.; investigation, D.D., W.D. and H.L.; resources, D.D., W.D. and H.L.; data curation, A.J. and H.L.; writing—original draft preparation, A.J., D.D. and W.D.; writing—review and editing, A.J. and H.L.; visualization, H.L.; supervision, H.L.; project administration, H.L.; funding acquisition, H.L. All authors have read and agreed to the published version of the manuscript.

**Funding:** The research reported was financially supported by the National Natural Science Foundation of China (No. 51908146), and the project features innovation of Guangdong Province (No. 2019KTSCX190).

**Data Availability Statement:** Data are contained within the article.

**Acknowledgments:** The authors would like to thank all partners of the School of Transportation and Civil Engineering & Architecture, Foshan University for their contribution to this research, with specific thanks to all partners in the China Construction Steel Structure Guangdong Corp. Ltd. Additionally, the authors thank all partners at the School of Civil Engineering, Southeast University, Nanjing who have been extremely cooperative in their involvement in this research.

**Conflicts of Interest:** The authors declare no conflict of interest.

## Appendix A. Coefficients $A_{11}$, $A_{12}$, $A_{13}$, $A_{21}$, $A_{22}$ and $A_{23}$

Coefficients $A_{11}$, $A_{12}$, $A_{13}$, $A_{21}$, $A_{22}$ and $A_{23}$ are given by

$$A_{11} = \frac{\left[6\zeta\Theta A_0 r_{x0}^2 a + I_m\left(1 - e^{-3a\Theta}\right)\right]R}{3aI_m r_{x0}^2 EA_0} \tag{A1}$$

$$A_{12} = \frac{\left[(3a\cos\Theta - \sin\Theta)e^{-3a\Theta} - 3a\right]R^2}{EAr_{x0}^2(9a^2+1)} + \frac{2\zeta\Theta R^2(1-\cos\Theta)}{EI_m} + \frac{\left(1-e^{-3a\Theta}\right)R^2}{3aEA_0 r_{x0}^2} \tag{A2}$$

$$A_{13} = qR^3\left\{\frac{\left[I_m\left(9a^2\cos c - 9a^2 - 1\right) - 6\Theta\zeta r_{x0}^2 a\left(3a^2+1\right)A\sin\Theta\sin c\right]e^{3ca} + I_m}{3ae^{3ca}EAr_{x0}^2(9a^2+1)I_m}\right\} \\ + \frac{qR^3\sin c(3a\sin\Theta + \cos\Theta)}{EAe^{3a\Theta}r_{x0}^2(9a^2+1)} \tag{A3}$$

$$A_{21} = \frac{\left[I_m(3a\cos\Theta - \sin\Theta)e^{-3a\Theta} - A\Theta\left(18a^2+2\right)\zeta r_{x0}^2\cos\Theta - 3aI_m\right]R^2}{EI_m(9a^2+1)Ar_{x0}^2} \tag{A4}$$

$$A_{22} = \frac{\left[ae^{-a\Theta}(1-\beta)\left(a\sin^2\Theta + \sin(2\Theta)\right) + \left(a^2+2\beta+2\right)\left(1-e^{-a\Theta}\right)\right]R}{EAa(a^2+4)} \\ + \frac{R^3}{E}\left[\frac{2\Theta\zeta\cos\Theta}{I_m}(\cos\Theta - 1) - \frac{9a^2\cos^2\Theta + 2 - 3a\sin(2\Theta)}{3e^{3a\Theta}r_{x0}^2 aA(9a^2+4)}\right] \\ + \frac{R^3}{EAr_{x0}^2}\left[\frac{(3a\cos\Theta - \sin\Theta)e^{-3a\Theta} - 3a}{9a^2+1} + \frac{9a^2+2}{3(9a^2+4)a}\right] \tag{A5}$$

and

$$A_{23} = \frac{qR^2\sin c}{EA}\left\{\frac{(1-\beta)[\sin(2\Theta)a + 2\cos(2\Theta)]}{2e^{a\Theta}(a^2+4)} - \frac{R^2[3\sin(2\Theta)a + 2\cos(2\Theta)]}{2e^{3a\Theta}(9a^2+4)r_{x0}^2}\right\} \\ + \frac{qR^2}{EA}\left\{\frac{\left[(2\beta-1)a^2 + 2\beta+2\right]\cos c + a\left(a^2\beta+\beta+3\right)\sin c}{e^{ca}a\left(a^4+5a^2+4\right)} - \frac{R^2\left[\cos c\left(9a^2-2\right) - 9a\sin c\right]}{3a\left(81a^4+45a^2+4\right)r_{x0}^2 e^{3ca}}\right\} \\ + \frac{qR^2}{E}\left\{\frac{R^2\Theta\zeta\sin c\sin(2\Theta)}{I_m} + \frac{\cos c}{Aa}\left[\frac{R^2\left(9a^2+2\right)}{3r_{x0}^2(9a^2+4)} - \frac{a^2+2\beta+2}{a^2+4}\right] + \frac{a}{A}\left[\frac{3R^2 r_{x0}^{-2}}{9a^2+1} + \frac{1}{a^2+1}\right]\right\} \tag{A6}$$

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
