# Peer review of "Out-Plane Buckling of Arches with Variable Cross-Section"

_buildings, doi:10.3390/buildings13112838_

Round 1

Reviewer 1 Report

Comments and Suggestions for Authors

The manuscript titled "Out-plane Buckling of Arches with Variable Cross Section" combines literature to discuss the buckling problem of uniform cross-section arches and briefly introduces the shortcomings of current research on the buckling problem of variable cross-section arches. The ANSYS arch rib model is established by the theoretical derivation of analytical solutions to carry out the study of buckling outside the variable cross-section arch plane. The logic of the manuscript is fluid, the issue is generally relevant to the journal, and the innovativeness is acceptable.The following suggestions are for reference.

1. In order to facilitate the analysis of the basic assumptions for out-of-plane buckling problems, it is necessary to clarify which assumptions are proposed by the author for the model in this study and which are the basic assumptions in basic mechanics. What impact will targeted assumptions have on model analysis? In other words, are these assumptions safe enough for engineering applications? Please explain. For example, the arch section is assumed to be rectangular. In the field of bridge engineering, such arch ribs are commonly used in deck arch bridges and through arch bridges with small and medium-sized spans. For half-through and through arch bridges with larger spans, steel tube concrete arch ribs are often used. Is this assumption applicable to other steel truss structures?

2. In the manuscript, the BEAM188 unit is used to establish the solid model of ANSYS arch ribs. The BEAM188 unit can intuitively display the internal force cloud map of the structure. Before comparing the finite element value and analytical solution of the internal force of the arch ribs, can the internal force distribution of the arch ribs under different working conditions be given?

3. Can the structural form of ANSYS arch ribs be changed by adjusting parameters during parameter analysis, such as the section height ratio? Also, calculate the finite element model, provide the stress cloud map of the arch rib, which enhances persuasiveness, and further compare and analyze the analytical solution and the finite element solution.

4. For the Castigliano theorem, there is no need to list the formulas and derivation processes known in books and literature. Please cite and simplify them, highlighting the main derivation and analysis process of this study.

Comments on the Quality of English Language

The language expression is fluent and the logic is clear. Please further modify some long sentences and use more academic expressions.

Author Response

Dear reviewer:

Thank you very much for your pertinent review suggestions on the manuscript "Out-plane Buckling of Arches with Variable Cross-section". The following are the improvements I have made to the paper:

Question 1:In order to facilitate the analysis of the basic assumptions for out-of-plane buckling problems, it is necessary to clarify which assumptions are proposed by the author for the model in this study and which are the basic assumptions in basic mechanics. What impact will targeted assumptions have on model analysis? In other words, are these assumptions safe enough for engineering applications? Please explain. For example, the arch section is assumed to be rectangular. In the field of bridge engineering, such arch ribs are commonly used in deck arch bridges and through arch bridges with small and medium-sized spans. For half-through and through arch bridges with larger spans, steel tube concrete arch ribs are often used. Is this assumption applicable to other steel truss structures?

Answer 1:Thank you very much for the expert advice. The assumptions of the paper are established to take into account the safety of engineering applications. Items (1), (2), (3) and (5) are the basic assumptions of fundamental mechanics, easy to calculate and have no significant impact on the results. Item (4) has been proposed in the model of this study. The rectangular shape can be applied to small and medium-sized arch Bridges with upper and lower bearings. However, it is also suitable for large span intermediate and underground arch Bridges with guaranteed stiffness, whether concrete steel pipe arch rib or other steel truss structures.(Lines 115-122)

Question 2: In the manuscript, the BEAM188 unit is used to establish the solid model of ANSYS arch ribs. The BEAM188 unit can intuitively display the internal force cloud map of the structure. Before comparing the finite element value and analytical solution of the internal force of the arch ribs, can the internal force distribution of the arch ribs under different working conditions be given?

Answer 2:Thank you very much for the expert advice. The internal force cloud image of the arch with variable cross section has been added in the article "5.1. Numerical model of the arch with variable cross section" as required.(lines 255-258)

Question 3:Can the structural form of ANSYS arch ribs be changed by adjusting parameters during parameter analysis, such as the section height ratio? Also, calculate the finite element model, provide the stress cloud map of the arch rib, which enhances persuasiveness, and further compare and analyze the analytical solution and the finite element solution.

Answer 3:Thank you very much for the expert advice. The stress cloud image of the arch with variable cross section has been added in the article "5.1. Numerical model of the arch with variable cross section" as required.(lines 259-261). In "5.2. Comparative analysis with finite element results", the analytical solution with added stress was compared with the finite element solution.(lines 263-283)

Question 4:For the Castigliano theorem, there is no need to list the formulas and derivation processes known in books and literature. Please cite and simplify them, highlighting the main derivation and analysis process of this study.

Answer 4:Thank you very much for the expert advice. Formulas and derivations in books on Castillano's theorem have been simplified as requested, and formulas (15) - (18) have been deleted.

Reviewer 2 Report

Comments and Suggestions for Authors

I have analysed this review article about buckling of arches.

The manuscript is indeed interesting in its scope and aims, it is also very scientific and important since we have relatively few reports about this field

I do not see any problems with English language.

The article is quite developed from the technical point of view and we do not learn a great deal about the technical feasibility of these systems, their adequacy or appropriateness in different types of stress-strain situations.

That said there are in my humble opinion no important shortcomings in this article. If only that the authors do not detail the way to obtain stresses in hyper-static arches nor they mention this difficult matter in the references. The analysed arch seems to be limited to steel construction but its concrete counterpart is not mentioned in the manuscript and it would be relevant for the readers. The drawings and graphs are well crafted and informative.

The analysis performed by ANSYS and other approximate methods is satisfactory.

There is a detailed mathematical analysis based on Castigliano’s principles.

I acknowledge that the authors have worked thoroughly on the matter but still the outcome is quite sufficient and convincing

The results of the manuscript is, therefore, totally consequential to me.

Summary of evaluation: The article is sufficiently developed and presents relevant properties in objectivity and technical matters.  It should be considered for publication if only with some minor polishing in the way expressed above.

Author Response

Dear reviewer:
Thank you very much for your pertinent review suggestions on the manuscript "Out-plane Buckling of Arches with Variable Cross-section", and thank you very much for your affirmation and support of my research content. I will continue to study the buckling problem of arches in depth, and strive to contribute my own strength to the development of the future arch structure.
Finally, I wish you good health and success in your career.

Reviewer 3 Report

Comments and Suggestions for Authors

The authors have developed an elastic analytical research of out-plane buckling of arches with variable cross-section under a uniformly distributed radial local load. The Castigliano's second theorem is used to establish pre-buckling force method equilibrium equations for variable cross-sectional arches under a uniformly distributed radial local load, and corresponding analytical solutions of normal stress, axial compression and the bending moment are obtained. Based on the energy method and the Ritz method, analytical solutions of the critical load for the elastic out-plane buckling of arches with variable cross-section are derived. Comparisons with ANSYS results implemented and indicated that the analytical solutions are able to accurately predict the pre-buckling internal forces and critical out-plane buckling load of variable cross-section arches subjected to a uniformly distributed radial local load.

In my opinion, the work is interesting and fits into the scope of journal. Hence, it can be accepted after following revisions:

1-      Pleases present the novelties and research highlights of your study in the last paragraph of introduction section.

2-      Please provide a brief review about the structure of paper at the end of Introduction.

3-      Te following researches about buckling analysis of rings and shells may be useful to be reviewed. For, example:

https://doi.org/10.1142/S0219455414500916

https://doi.org/10.1007/s00419-022-02132-2

4-      Please present details of mesh convergence studies for verification example in Ansys.

5-      What is the applied BC in Ansys?

6-      The authors must investigate the effect different BCs. The authors just have only considered one BC.

7-      What are the superiorities of present study over commercial FE softwares, since the authors have verified their problem by FEM results? The problem could also be solved by FEM softwares without any difficulties. The authors must clearly discuss about this point to show the novelties of their study.

Author Response

Dear reviewer:

Thank you very much for your pertinent review suggestions on the manuscript "Out-plane Buckling of Arches with Variable Cross-section". The following are the improvements I have made to the paper:

Question 1: Pleases present the novelties and research highlights of your study in the last paragraph of introduction section.

Answer 1:Thank you very much for the expert advice. As requested, the highlights of this study are described in the last part of the introduction. In this paper, the variable section form of exponential function based on natural constant e is adopted, which plays a great role in simple calculation. In this paper, arches with continuous and uniform changes in height are studied, but there is a lack of out-of-plane buckling elasticity analysis for these arches. The load form in this paper is the uniformly distributed radial local load, which is more close to the actual project.(Lines 77-91)

Question 2: Please provide a brief review about the structure of paper at the end of Introduction.

Answer 2: Thank you very much for the expert advice. In accordance with the requirements, the last paragraph of the introduction has been sorted out the context of the paper. In this paper, the normal stress, axial compression and bending moment of a variable section arch before buckling are solved by combining the Timoshenko beam theory and Castigliano's second theorem. On the basis of accurate axial compression and bending moment, the analytical solution of the critical out-plane buckling load of the arch with variable section is derived, and the numerical model is established by ANSYS software to verify the accuracy of the analytical solution. Finally, a certain amount of parametric analysis of internal force and out-plane critical buckling load is carried out to demonstrate the excellent mechanical properties of reasonably designed variable section arch.(Lines 77-91)

Question 3:Te following researches about buckling analysis of rings and shells may be useful to be reviewed. For, example:

https://doi.org/10.1142/S0219455414500916

https://doi.org/10.1007/s00419-022-02132-2

Answer 3:Thank you very much for the expert advice. Through reading the above two papers of great reference value, I have benefited greatly from the writing of the introduction. (Reference [31] and reference [32] are cited in the above two papers.)

Question 4: Please present details of mesh convergence studies for verification example in Ansys.

Answer 4:Thank you for your professional advice. In this paper, the ANSYS software is used, employing the beam188 element for modeling the variable-section arch. By adjusting the section height of each beam188 element in the arch finite element model, the model is modified to achieve a continuously varying section based on the natural constant e. Additionally, the beam188 element is a one-dimensional beam element that considers shear deformation. For the analysis in this paper, it only involves buckling eigenvalue analysis and static linear analysis. Therefore, there is no mesh convergence issue related to two-dimensional shell elements or three-dimensional solid elements.

Question 5:What is the applied BC in Ansys?

Answer 5:Thank you for your professional advice. In this paper, the variable-section arch employs in-plane elastic rotational constraint boundary conditions and out-plane pin-ended boundary conditions, which are implemented in ANSYS simulation using the combine14 element.

Question 6:The authors must investigate the effect different BCs. The authors just have only considered one BC.

Answer 6:Thank you for your professional advice. In this paper, when the stiffness of the in-plane elastic rotational constraint is equal to zero, it represents in-plane pin-ended boundary conditions. When the flexibility of the in-plane elastic rotational constraint is equal to zero, it represents in-plane fixed boundary conditions. The variation of different boundary conditions has an impact on the critical load. Please refer to Figure 16 for details.

Question 7:What are the superiorities of present study over commercial FE softwares, since the authors have verified their problem by FEM results? The problem could also be solved by FEM softwares without any difficulties. The authors must clearly discuss about this point to show the novelties of their study.

Answer 7:Thank you very much for your professional advice. The advantages of the analytical method include:

  1. The critical load of the arch structure can be calculated through formulas with only the dimensional information of the arch, without the need for cumbersome finite element modeling.
  2. The analytical method facilitates optimization design analysis for the structural calculations. This is because optimizing the design of arch structures with respect to dimensions requires hundreds or even thousands of calculations of critical loads. It would be impractical to establish corresponding finite element models for each arch structure.(Lines 149-153)

Round 2

Reviewer 3 Report

Comments and Suggestions for Authors

The paper is revised accordingly and can be accepted in the present form.